# A Resilient Method for Visual–Inertial Fusion Based on Covariance Tuning

**DOI:** 10.3390/s22249836

**Published:** 2022-12-14

**Authors:** Kailin Li, Jiansheng Li, Ancheng Wang, Haolong Luo, Xueqiang Li, Zidi Yang

**Affiliations:** Institute of Geospatial Information, Information Engineering University, Zhengzhou 450001, China

**Keywords:** resilient sensor fusion, simultaneous localization and mapping, visual–inertial fusion, nonlinear optimization, covariance tuning

## Abstract

To improve localization and pose precision of visual–inertial simultaneous localization and mapping (viSLAM) in complex scenarios, it is necessary to tune the weights of the visual and inertial inputs during sensor fusion. To this end, we propose a resilient viSLAM algorithm based on covariance tuning. During back-end optimization of the viSLAM process, the unit-weight root-mean-square error (RMSE) of the visual reprojection and IMU preintegration in each optimization is computed to construct a covariance tuning function, producing a new covariance matrix. This is used to perform another round of nonlinear optimization, effectively improving pose and localization precision without closed-loop detection. In the validation experiment, our algorithm outperformed the OKVIS, R-VIO, and VINS-Mono open-source viSLAM frameworks in pose and localization precision on the EuRoc dataset, at all difficulty levels.

## 1. Introduction

Due to the maturation of positioning navigation and time (PNT) systems [1] and elastic PNT frameworks [2], multisource PNT data fusion techniques are becoming increasingly intelligent and adaptive. Visual–inertial navigation systems (VINS) are commonly used to combine multisource PNT data, and they are commonly used in mobile equipment, mobile robots, and small flying devices for simultaneous localization and mapping (SLAM). The development of a robust visual–inertial fusion algorithm for pose and location determination would represent a significant step forward for the use of intelligent vehicles in complex or dynamic environments.

Due to the camera′s characteristic of small size, low cost, low power consumption and easy assembly, the visual simultaneous localization and mapping system (VSLAM) has attracted wide attention [3]. As an important branch of the VSLAM system, visual odometry (VO) is widely studied, such as library for visual odometry (LIBVISO) [4], semi-direct monocular visual odometry (SVO) [5], and direct sparse odometry (DSO) [6] which are the three most representative visual odometry algorithms. Mainstream VSLAM systems can be divided into two categories. The first one is filtering-based methods, such as Mono-SLAM [7]. The second one is the optimization methods of bundle adjustment (BA), such as parallel tracking and mapping (PTAM) [8]. Since Strasdat [9] pointed out that optimization algorithms are more cost-effective than filtering algorithms, BA-based VSLAM algorithms have developed rapidly, such as ORB-SLAM [10] with sparse features and LSD-SLAM [11] with dense features. These algorithms achieve better relative accuracy in common indoor scenes. However, the VSLAM algorithm inevitably has the disadvantages of difficult scale estimation (monocular camera), strong scene texture dependence, great influence by illumination, and is extremely unstable under high dynamics. The high-precision angular velocity and acceleration measurement of inertial sensors in a short time can make up for the shortcomings of the VSLAM system and improve the accuracy and stability of the system [12]. Recently, visual–inertial SLAM has become an important area of study in SLAM research. In viSLAM, the visual sensor front end extracts features directly from sensor-captured images or through visual features. The high short-term precision of inertial measurement units (IMUs) is then used to constrain the visual data, which results in precise localization. The development of filter-based and factor graph optimization (FGO)-based sensor fusion techniques led to the emergence of several highly performant open-source visual–inertial frameworks. One example of a filter-based open-source framework is the multi-state constraint Kalman filter (MSCKF) proposed by Mourikis in 2007 [13]. MSCKF treats all poses in the active window as filter state variables, and the state vector does not include 3D feature positions, which reduces computational complexity to some extent. However, the local state estimates of the MSCKF algorithm tend to be imprecise, and the problem of inconsistent filtering estimation exist, which makes the unobservable state produce false observability, such as heading angle. Subsequently, the visual–inertial integrated method based on MSCKF extended SR-ISWF [14,15] and unscented Kalman filter (UKF) algorithm [16] were proposed. Other open-source frameworks, such as the extended Kalman filter (EKF)-based ROVIO (2015) fusion framework [17], MSCKF-based R-VIO (2018) framework [18], and OpenVINS (2020) [19] were subsequently proposed. In ROVIO, the sparse direct method is used as the visual front end, and a photometric residual is employed in the EKF update; although these innovations improved localization precision, the ROVIO framework is computationally complex. R-VIO is a robocentric viSLAM framework that reformulates VINS with respect to a moving local frame, whereas OpenVINS is an MSCKF-based framework that improves on MSCKF in terms of landmark estimation; both are highly efficient and lightweight visual–inertial frameworks. Filter-based viSLAM frameworks tend to produce insufficiently precise landmark coordinates, and they are, in theory, highly susceptible to linear errors, which results in suboptimal robustness and precision in complex settings. FGO-based viSLAM frameworks, such as iSAM2 [20], VIM-SLAM [21], OKVIS [22], VINS-Mono [23], and ORB-SLAM3 [24], have seen rapid development since the emergence of the IMU preintegration technique [25] and the refinement and validation of preintegration theory by Forster et al. [26]. The iSAM2 uses Bayes tree to achieve incremental smoothing and mapping, which takes full advantage of the sparsity of the Hessian matrix. VIM-SLAM is an extension algorithm of ORB-SLAM that incorporates inertial sensors. OKVIS processes landmarks using a keyframe-based sliding window, and it incorporates weighted IMU preintegration and reprojection error terms into a factor graph for nonlinear optimization with marginalization. VINS-Mono is a monocular visual–inertial state estimator that utilizes a loop detection module for re-localization and also performs four-degree-of-freedom pose graph optimization to enforce global consistency. This approach improves trajectory fitting and estimator initialization compared to its contemporaries while remaining computationally efficient. ORB-SLAM3 uses maximum-a-posteriori (MAP) estimation from the initialization phase, which allows for rapid initialization, and also provides multi-map data association to facilitate loop detection and BA; these features allow ORB-SLAM3 to operate robustly for long periods with poor visual information.

Although the aforementioned visual front end, loop detection, and initialization methods have greatly improved the robustness of viSLAM, they all share the same flaw: although filter-based visual–inertial odometry is able to propagate and update covariance matrices, the residual introduced to each visual measurement is always a fixed a priori value. For example, R-VIO always assumes that all landmarks have a tracking precision of 1 pixel. Although filtering-based algorithms such as ROVIO and MSCKF update the covariance matrix, the covariance introduced by the newly added sensor′s measurement of filter are fixed. Apparently, the initial values of these covariance are independent of the propagation of the covariance. In FGO-based viSLAM, it is necessary to determine the information matrix of the sensor outputs, i.e., the inverse of the covariance matrix, prior to nonlinear optimization. For IMUs, the information matrix can be derived from the propagation of the IMU preintegration, which is initialized using the random walk error and noise of the IMU. For visual sensors, the magnitude of the covariance matrix elements often depends on the image registration precision of the landmarks; as this parameter roughly describes the reprojection error, it is a measure of the visual sensors’ precision. Many open-source viSLAM frameworks employ quantitatively descriptive methods to determine the information matrix of the visual sensors’ outputs. For instance, VINS-Mono assumes that landmarks have a registration precision of 1.5 pixels. In ORB-SLAM3, the information matrix of the reprojection error is hierarchically defined by a pyramidal structure, albeit with each layer having fixed values; the first layer has a reprojection error of 1 pixel, and the reprojection errors of the subsequent layers are set according to some ratio. A fixed covariance matrix based on empirical a priori values is a simple and feasible strategy. However, this implies that the visual weights are fixed, which may result in a loss of pose and positioning precision in complex or dynamic scenarios in which the actual sensor outputs may not be compatible with the pre-determined ratio of visual and inertial weights. In the course of our experiments on VINS-Mono test, we found that in some complex scenes, by changing the registration precision of 1.5 pixels to 0.5 pixels, the accuracy increased by about 30%, which led to the study in this paper. Weight ratio tuning problem is common in the field of surveying and mapping. Yang et al. [27] proposed an adaptive filtering algorithm that estimates and amends model and noise characteristics, with robust estimation [28] and variance component estimation used to solve the weight adjustment problem [29]. However, the iterative calculations required for variance component estimation are computationally complex and intractable for real-time applications like SLAM. As a result, this approach is not commonly used in SLAM.

To address the aforementioned problems, we propose a method for visual–inertial fusion based on covariance tuning. During this study, we analyzed the error patterns of visual sensors and IMUs, and studied strategies to adjust the covariance of the visual sensors’ outputs. On this basis, we chose an approach in which the unit-weight root-mean-square errors (RMSEs) are computed from a posteriori visual reprojection errors and then used to construct a covariance tuning function for some interval. We also introduced a re-optimization process to the cycle in which the new visual covariance are used for nonlinear optimization to produce new pose solutions. In validation tests conducted using the time-synchronous EuRoc dataset [30], our method outperformed VINS-Mono by 2.39–35.65% in pose precision without closed-loop detection.

## 2. Methods and Principles

Figure 1 illustrates the architecture of the proposed covariance tuning-based method for resilient sensor fusion in visual–inertial odometry (VIO). This architecture was inspired by VINS-Mono. The architecture of the proposed algorithm consists of five modules: data acquisition and processing, data management, VIO initialization, back end, and re-optimization. The data acquisition and processing module receives and preprocesses the measurement data of the camera and IMU; the data management module manages IMU preintegration and features information in images; VIO initialization estimates gravity vector, velocity, gyroscope bias, and metric scale to ensure the normal operation of the system; the back end performs nonlinear optimization on the sliding window composed of multiple image frames; the re-optimization module uses the covariance tuning model to re-optimize the factor graph constructed at the back end to obtain more accurate navigation results. The architecture of the proposed method retains the same visual front end as VINS-Mono. The raw data inputs of the system include camera-captured images, gyroscope-measured angular velocities, and acceleration values from the accelerometer. The visual front end uses the Good Features To Track (GFTT) proposed by Shi-Tomasi in combination with the Lucas–Kanade optical flow method [31,32], as this strategy provides excellent real-time performance and sensitivity to movement. An IMU preintegration model is used to process the IMU measurements. VIO initialization is performed using the fast VIO initialization method described by Qin and Shen [33]. First, visual measurements are used to perform structure from motion (SfM) for a short period of time. The core work is to determine the three-dimensional coordinates in the camera coordinate system by triangulating the features detected in the images over a period of time, and use the efficient perspective-n-point (EPnP) [34] method to solve the camera poses. IMU preintegration is then performed to roughly align the visual and IMU measurements to estimate the initial state of the system (i.e., gravity vector, velocity, gyroscope bias, and metric scale). This ensures that the VIO system will operate with high-quality initial states. After the system has been initialized, nonlinear optimization is performed using a keyframe-based sliding window to construct a factor graph model, which consists of factor nodes such as the sliding window’s state nodes and visual reprojection factors, IMU preintegration factors, and marginalization factors. Each factor represents the residual of the error equation established by the relative sensor. The marginalization factor is a constraint brought by fixing some old states when the sliding window is updated. It is obtained by the Schur complement operation of the previous sensor factor. The specific process is clearly described in the reference [35]. Nonlinear optimization in the back end is an iterative solution of the error equation in the factor graph. After the first round of nonlinear optimization, the unit-weight RMSEs of the reprojection residuals will be used as inputs for the covariance tuning model, which adjusts the visual covariance. In the subsequent re-optimization process, the new visual covariance matrix is used to reconstruct the factor graph model for a second round of nonlinear optimization, which ultimately solves for the pose and localization of the system. As the proposed method is meant to ensure pose and localization precision in real-time navigation, we only evaluated the forward pose estimation algorithm and did not consider global pose recalibration by loop detection.

Here we briefly describe the variables and frames used in this work. (·)*^b^* represents the IMU frame, which is also referred to as the carrier frame. (·)*^w^* represents the world frame, which is obtained by leveling the initial carrier frame. (·)*^c^* is the camera frame, whose direction is defined by its super- and subscripts. For instance, ${\left(\cdot \right)}_{b}^{w}$ represents the transformation of the carrier frame to the world frame, and the superscript $\hat{\cdot }$ is the measured value. Rotation is represented by the rotational matrix ***R*** and rotation quaternion **q**, and the parameters being optimized by the nonlinear optimization algorithm are quaternions.

### 2.1. Construction of the Factor Graph

In the proposed algorithm, the length of the sliding window is marginalized to avoid redundant optimization and improve computational efficiency. The image frames in the sliding window consist of the current image frame and previously found keyframes, and the to-be-optimized state vectors are ascertained based on the keyframes and the current image frame. All of the state vectors that await optimization in the sliding window are defined as follows:(1)$$\begin{array}{l} X={\left[x_{1}{}^{T},x_{2}{}^{T},\ldots ,x_{n}{}^{T},p_{c}^{b},q_{c}^{b},\rho_{1},\rho_{2},\ldots ,\rho_{m}\right]}^{T}\\ x_{k}={\left[p_{b_{k}}^{w},v_{b_{k}}^{w},q_{b_{k}}^{w},b_{a},b_{\omega }\right]}^{T},k\in \left[0,1,\ldots ,n\right] \end{array}$$

In this equation, *n* and *m* are the number of to-be-optimized state vectors in the sliding window and the number of observed features, respectively. $p_{c}^{b}$ and $q_{c}^{b}$ are the translation and rotation quaternia between the camera and carrier frames, respectively. *ρ* is the inverse depth of the feature, which represents its metric scale.$\;p_{b_{k}}^{w}$ and $q_{b_{k}}^{w}$ are the translation and rotation quaternia, respectively, of the *k*th-state vector between the camera and carrier frames, and $v_{b_{k}}^{w}$ is the carrier velocity in the world frame that corresponds to the *k*th-state vector. **b***_a_* and **b***_ω_* are the biases of the IMU’s accelerometer and gyroscope, respectively.

The observation model of the algorithm is shown in Figure 2. Within the sliding window, the movement of the carrier is represented by the camera’s sampling frequency. As there will be some degree of overlap between adjacent frames, the same feature may be observed in multiple images. As the carrier continues to move, a continuous stream of new frames is selected as keyframe *x_k_*. If the second-newest frame is selected as a keyframe, the oldest keyframe in the sliding window *x*_1_ is marginalized. It should be noted that IMU measurements of some duration will also be taken between adjacent image frames. Figure 3 illustrates a factor graph model that corresponds to a window length of 11 (*n* = 11). The preintegration factor *b_k_* is constructed from IMU measurements that occur between state variables in the sliding window, and the visual reprojection factor *c_j_* is formed by image frames that observe the same feature. As the sliding window moves, a new image frame enters the window as an old state is marginalized (i.e., its state vector is fixed). Nonetheless, the state vectors in the new sliding window are constrained by the state vector of this old keyframe. These constraints include IMU preintegration constraints between the marginalized image frame and the current oldest image frame in the new sliding window, and reprojection error constraints with the marginalized image frame being the first observation image frame and the constraints from the previous marginalization. All of these constraints are added to the nonlinear constraints as the sliding window’s marginalization factor.

The overall cost function of the nonlinear optimization may be constructed from the factor graph model of the sliding window:(2)$$f(X)=\underset{X}{\min}\left\{{\left\|r_{p}-H_{p}X\right\|}^{2}+\sum_{k\in B}{\left\|r_{B}\left({\hat{z}}_{b_{k+1}}^{b_{k}},X\right)\right\|}_{P_{b_{k+1}}^{b_{k}}}^{2}+\sum_{(l,j)\in C}{\left\|r_{C}\left({\hat{z}}_{l}^{c_{j}},X\right)\right\|}_{P_{l}^{c_{j}}}^{2}\right\}$$

In this equation, *r_p_* and ***H****_p_* are the a priori marginalization information, i.e., the marginalization residual and the Hessian matrix of the marginalization, respectively; $r_{B}\left({\hat{z}}_{b_{k+1}}^{b_{k}},X\right)$ is the IMU preintegration factor; $r_{C}\left({\hat{z}}_{l}^{c_{j}},X\right)$ is the visual reprojection factor; $\hat{z}$ is a measured value; *B* is a set formed by the series of IMU preintegrations; *C* is a set formed by the image frame series and all features; $P_{b_{k+1}}^{b_{k}}$ is the covariance of the preintegration noise term formed by the *k*th and *k* + 1th carrier state variables, and $P_{l}^{c_{j}}$ is the covariance of the noise in visual observations.

When performing iterative optimization using the Newton–Raphson method, the incremental equation that corresponds to the overall cost function (Equation (2)) is:(3)$$\begin{array}{l} \left(H_{p}+\sum_{k\in B}J_{b_{k+1}}^{b_{k}}{}^{T}P_{b_{k+1}}^{b_{k}}{}^{-1}J_{b_{k+1}}^{b_{k}}+\sum_{(i,j)\in C}J_{l}^{c_{j}}{}^{T}P_{l}^{c_{j}}{}^{-1}J_{l}^{c_{j}}\right)\Delta X\\ =b_{p}+\sum_{k\in B}J_{b_{k+1}}^{b_{k}}{}^{T}P_{b_{k+1}}^{b_{k}}{}^{-1}r_{B}+\sum_{(i,j)\in C}J_{l}^{c_{j}}{}^{T}P_{l}^{c_{j}}{}^{-1}r_{C} \end{array}$$

In this equation, *J* is the Jacobian, ***b****_p_* is the marginalization constant, and Δ**X** is the increment of the variable to be optimized.

It may be observed from the incremental equation that the iterative optimization is affected by the covariance matrix. If the visual reprojection covariance $P_{l}^{c_{j}}$ is large, information matrix term $P_{l}^{c_{j}}{}^{-1}$ will be small, which reduces its effect on the incremental equation, and increases the influence of the observed IMU preintegration term. The opposite applies if $P_{l}^{c_{j}}$ is small. Therefore, the relative magnitudes of the visual and IMU covariance will affect the final result of the optimization.

### 2.2. Visual Reprojection Factor

Based on the pinhole camera model, we used the definition proposed by Tong et al. [23] for visual residuals to define the reprojection error of the visual front end in terms of a unit sphere. The advantage of this approach is that wide-angle and fish-eye lenses may be modeled using unit rays connected to a unit sphere. Suppose that feature *l* is first observed in image series *i*. The reprojection error that occurs when image series *j* revisits feature *l* can then be expressed as follows:(4)$$\begin{array}{l} r_{c_{i}}^{c_{j}}=\left({\hat{p}}_{l}^{c_{j}}-\frac{p_{l}^{cj}}{\left\|p_{l}^{cj}\right\|}\right)\\ {\hat{p}}_{l}^{c_{j}}=\pi_{c}{}^{-1}\left(\left[\begin{array}{l} {\hat{u}}_{l}^{c_{j}}\\ {\hat{v}}_{l}^{c_{j}} \end{array}\right]\right)\\ p_{l}^{c_{j}}=R_{b}^{c}\left(R_{w}^{b_{j}}\left(R_{b_{i}}^{w}\left(R_{c}^{b}\frac{1}{\rho_{l}}\pi_{c}{}^{-1}\left(\left[\begin{array}{l} {\hat{u}}_{l}^{c_{i}}\\ {\hat{v}}_{l}^{c_{i}} \end{array}\right]\right)+p_{c}^{b}\right)+p_{b_{i}}^{w}-p_{b_{j}}^{w}\right)-p_{c}^{b}\right) \end{array}$$

In this equation, $\left[{\hat{u}}_{l}^{c_{i}},{\hat{v}}_{l}^{c_{i}}\right]$ are the image-plane coordinates of *l* (in pixels) in image series *i*; $\left[{\hat{u}}_{l}^{c_{j}},{\hat{v}}_{l}^{c_{j}}\right]$ are the image-plane coordinates of *l* in image series *j*; $R_{b}^{c}$ and $P_{c}^{b}$ are the rotation matrix and translation that relate the camera and carrier frames, respectively, with the direction of rotation/translation indicated by their super- and subscripts. $R_{w}^{b}$ and $P_{b}^{w}$ are the rotation matrix and translation that relate the carrier and world frames, respectively, with the direction of rotation/translation indicated by their super- and subscripts. $\pi_{c}{}^{-1}\left(\cdot \right)$ is the pixel plane-to-unit vector transformation function, and *ρ_l_* is the inverse depth of the feature, which represents the scalar relationship between unit sphere vectors and real-world spatial coordinates.

In a pinhole camera, the scaling of the normalization plane to the unit sphere may be embedded in the estimation of *λ_l_*, the inverse depth of feature *l*. Therefore, $\pi_{c}{}^{-1}\left(\cdot \right)$ may be expressed as the internal parameter transform from pixel coordinates to normalization plane coordinates:(5)$$\begin{array}{l} \pi_{c}{}^{-1}\left(\left[\begin{array}{l} u\\ v \end{array}\right]\right)=K^{-1}\left[\begin{array}{l} u\\ v \end{array}\right]\\ K=\left[\begin{array}{ccc} f_{x} & 0 & u_{0}\\ 0 & f_{y} & v_{0}\\ 0 & 0 & 1 \end{array}\right] \end{array}$$

In this equation, [*u*, *v*] are the pixel coordinates of the point, [u_0_, v_0_] are the coordinates of the principal point, and f*_x_* and f*_y_* are the focal lengths of the image frame in the *x* and *y* directions, respectively.

The visual reprojection error factor can be expressed as follows:(6)$$r_{C}({\hat{z}}_{l}{}^{c_{j}},X)={\left[e_{1},e_{2}\right]}^{T}\cdot r_{c_{i}}^{c_{j}}$$

In this equation, **e**_1_ and **e**_2_ are orthogonal bases that span the unit sphere’s tangent plane.

### 2.3. IMU Preintegration Factor

As the carrier frame is identical to the IMU frame, the IMU’s gyroscope and accelerometer models may be expressed as follows:(7)$$\begin{array}{l} {\hat{\omega }}^{b}=\omega^{b}+b_{\omega }+n_{\omega }\\ {\hat{a}}^{b}=a^{b}+R_{w}^{b}g^{w}+b_{a}+n_{a} \end{array}$$

In these equations, $\hat{\omega }$ and $\hat{a}$ are the values measured by the gyroscope and accelerometer, respectively, and **ω** and **a** are the true angular velocity and acceleration, respectively. $R_{w}^{b}$ is the rotation matrix that relates the world frame to the carrier frame, and **g***^w^* is the gravitational acceleration of the world frame. **n***_ω_* and **n***_a_* are the additive noise of the gyroscope and accelerometer, which are modeled as Gaussian white noise, such that $n_{a}∼Ν\left(0,\sigma_{a}^{2}\right)$ and $n_{\omega }∼Ν\left(0,\sigma_{\omega }^{2}\right)$, respectively. **b***_ω_* and **b***_a_* are the gyroscope and accelerometer biases, respectively, which are modeled as Gaussian random walks ${\dot{b}}_{a}∼Ν\left(0,\sigma_{b_{a}}^{2}\right)$ and ${\dot{b}}_{\omega }∼Ν\left(0,\sigma_{b_{\omega }}^{2}\right)$, respectively.

As the IMU preintegrations are continuous in time, let us suppose that there is some reference time *i*. The position, velocity, and pose at *j* may then be expressed as:(8)$$\begin{array}{l} p_{b_{j}}^{w}=p_{b_{i}}^{w}+v_{b_{i}}^{w}\Delta t-\frac{1}{2}g^{w}\Delta t^{2}+R_{b_{i}}^{w}\iint_{t\in \left(i,j\right)}\left(R_{b_{t}}^{b_{i}}a^{b_{t}}\right)\delta t^{2}\\ v_{b_{j}}^{w}=v_{b_{i}}^{w}-g^{w}\Delta t+R_{b_{i}}^{w}\int_{t\in \left(i,j\right)}\left(R_{b_{t}}^{b_{i}}a^{b_{t}}\right)\delta t\\ q_{b_{j}}^{w}=q_{b_{i}}^{w}⊗\int_{t\in \left(i,j\right)}q_{b_{t}}^{b_{i}}⊗\left[\begin{array}{c} 0\\ \frac{1}{2}\omega^{b_{t}} \end{array}\right]\delta t \end{array}$$

In these equations, ⊗ indicates a quaternion multiplication.

The position, velocity, and pose preintegrations are defined as follows:(9)$$\begin{array}{l} \alpha_{ij}=\iint_{t\in \left(i,j\right)}\left(R_{b_{t}}^{b_{i}}a^{b_{t}}\right)\delta t^{2}\\ \beta_{ij}=\int_{t\in \left(i,j\right)}\left(R_{b_{t}}^{b_{i}}a^{b_{t}}\right)\delta t\\ \gamma_{ij}=\int_{t\in \left(i,j\right)}q_{b_{t}}^{b_{i}}⊗\left[\begin{array}{c} 0\\ \frac{1}{2}\omega^{b_{t}} \end{array}\right]\delta t \end{array}$$

In the algorithm, the IMU preintegrations are discretized by the median and then propagated. Based on the two equations above, the IMU preintegration factor and IMU preintegration error may be defined as follows:(10)$$r_{B}\left({\hat{z}}_{b_{k+1}}^{b_{k}},X\right)={\left[\begin{array}{l} r_{p}\\ r_{q}\\ r_{v}\\ r_{b_{a}}\\ r_{b_{g}} \end{array}\right]}_{15\times 1}=\left[\begin{array}{c} q_{w}^{b_{i}}\left(p_{b_{j}}^{w}-p_{b_{i}}^{w}-v_{b_{i}}^{w}\Delta t+\frac{1}{2}g^{w}\Delta t^{2}\right)-\alpha_{ij}\\ 2{\left[\gamma_{ij}⊗\left(q_{b_{i}}^{w}⊗q_{b_{j}}^{w}\right)\right]}_{xyz}\\ q_{w}^{b_{i}}\left(v_{b_{j}}^{w}-v_{b_{i}}^{w}+g^{w}\Delta t\right)-\beta_{ij}\\ b_{aj}-b_{ai}\\ b_{gj}-b_{gi} \end{array}\right]$$
where the *r* terms are residuals. ${\left[\cdot \right]}_{xyz}$ indicates that only the three-dimensional vector from the quaternion’s imaginary part will be taken.

### 2.4. Covariance Tuning Based on Unit-Weight RMSE

In many open-source viSLAM frameworks, the visual covariance is some fixed value that does not change after it is incorporated into the nonlinear optimization. For instance, in VINS-Mono, the covariance factor *σ* in the covariance matrix $P_{l}^{c_{j}}$ is a fixed value, 1.5/f, where f is the virtual focal length. Therefore, the reprojection error is assumed to be approximately 1.5 pixels in the image plane, which causes the visual weights to be fixed in any scenario. Although 1.5 pixels is a stable empirical threshold, σ = 1.5/f is a poor assumption if any significant change occurs in the sensor devices (e.g., if a different sensor model is used) or in the external environment. Therefore, a resilient and tunable scheme should be used to tune the visual and inertial weights to improve pose and localization precision.

The idea of using covariance tuning function to deal with SLAM problem was inspired by reference [29]. Yang′s application of variance component estimation has great implications for the covariance tuning model proposed in this paper. In reference [29], the unit-weight RMSE was used to estimate the variance component. Simulation results show that the variance component estimation method based on unit-weight RMSE can greatly improve the positioning accuracy of multi-sensor fusion. In order to ensure the real-time performance, with more complex establishment of sensor measurement in the field of SLAM, the variance component estimation with multiple iterative optimization will bring computational challenges. Although this method cannot be used directly, the idea of changing the measurement covariance of sensors by the posterior unit-weight RMSE is applicable. As the factor graph problem is a nonlinear least squares problem, if there are *m* sets of visual measurements, the visual residual after the first round of visual–inertial fusion and optimization may be expressed as follows:(11)$$V_{C}={\left[\begin{array}{c} r_{u_{1}},r_{v_{1}},\cdots ,r_{u_{m}},r_{v_{m}} \end{array}\right]}_{1\times 2m}^{T}$$

The unit-weight RMSE of each visual reprojection residual is given by the following:(12)$$\begin{array}{l} \sigma =\sqrt{\frac{V_{C}{}^{T}P^{-1}V_{C}}{n_{d}-t}}\\ P^{-1}={\left[\begin{array}{ccc} 1 & \cdots & 0\\ \vdots & \ddots & \vdots \\ 0 & \cdots & 1 \end{array}\right]}_{2m\times 2m}\\ n_{d}=2m\\ t=m+6 \end{array}$$
where ***P***^−1^ is the weight matrix, in unit weights; *n_d_* is the total number of dimensions of the visual reprojection factor, with each observation point providing two constraint equations, and *t* is the number of variables to be optimized, which includes *m* inverse depths and pose in six degrees of freedom.

The unit-weight RMSE of the visual reprojections is a reflection of visual quality. A large unit-weight RMSE indicates that the environment or carrier’s movements severely affect localization precision and that the visual measurements are of poor quality. On this basis, we propose a covariance-tuning function based on the unit-weight RMSE:(13)$$\begin{array}{l} P_{l}^{c_{j}}=\left[\begin{array}{cc} {\sigma^{\prime }}^{2} & 0\\ 0 & {\sigma^{\prime }}^{2} \end{array}\right],j\in \left(1,2,\cdots ,m\right)\\ \sigma^{\prime }=Q\left(\sigma^{2}\right)=\frac{k\sqrt{\sigma^{2}}}{\mathrm{lg}\left(\frac{1}{\sigma^{2}}\right)} \end{array}$$
where *σ*’ is the new covariance factor, *σ* is the unit-weight RMSE computed from the a posteriori residuals, *k* is the confidence factor, and lg(·) is the log function with base 10.

The inclusion of the log function is meant to increase the function’s sensitivity to change, and *k* = 2 corresponds to a confidence level of 95.44%. The function is plotted in Figure 4, which shows that the plot with the log function has larger gradients than the original plot, which increases the sensitivity of the re-optimization process to unit-weight RMSE.

The inputs of the covariance tuning function are the a posteriori unit-weight RMSEs of the visual reprojection residuals, which reflect the quality of the visual measurements. Using this parameter to adjust the covariance matrix, it becomes possible to accurately assign weights to the visual and inertial measurements and make the best use of the precision of each sensor.

### 2.5. Re-Optimization

Passing the unit-weight RMSEs to the covariance-tuning function produces a new visual covariance matrix that reflects the level of visual measurement noise after the first round of nonlinear optimization. During re-optimization, the new information matrix is determined according to the level of visual noise, which is then re-substituted into the factor graph for a second round of nonlinear optimization. Creating a new information matrix permits optimization of the ratio of weights between the visual and inertial measurements, which changes the reliance of the system on these measurements.

It should be noted that the weight of the marginalization factor will not be actively altered during re-optimization. As shown in Figure 5, the algorithm will use the new visual covariance matrix for marginalization. As the marginalization factor of the current sliding-window optimization was originally propagated from the marginalization process that occurred after the previous round of sliding-window optimization, the reprojection-residual covariance matrix would have been altered by the algorithm during the processing of the previous image frame. Therefore, when the marginalization factor is propagated to the nonlinear optimization of the current image frame, it is unnecessary to reweigh the visual reprojection factor in the marginalization process.

## 3. Experiments and Analysis

The experimental dataset was the EuRoc dataset, which consists of hardware-synchronized stereo camera data and IMU data from a hexacopter unmanned aerial vehicle (UAV). The ground-true pose values were acquired by a Vicon motion capture system and Leica laser tracker, which have millimeter-level precision. The EuRoc dataset includes two settings: a machine hall and an ordinary room (“Vicon Room”) in ETH Zurich. This dataset also includes 11 image series that were officially classified as “easy,” “medium,” and “difficult,” as shown in Table 1. We use “E” to represent “easy,” “M” to represent “medium,” “D” to represent “difficult,” and “/“ is used to separate different data segments in the same series.

### 3.1. Effectiveness of Covariance Tuning

First, the covariance factor σ for the to-be-optimized visual factor was set to 1.5/f, without further re-optimization. The V102 dataset from the Vicon Room was analyzed with this setting. The algorithm processes the image by frame, so the abscissa “Frame number“ represents the image frame sequence of the data segment V102, and the “Frame number“ in the subsequent figures also represents the same meaning. The results are shown in Figure 6. Most of the unit-weight RMSEs of the visual reprojection are within 10^−4^, and the visual reprojection factor is rather unstable. This is caused by changes in environmental illumination and the movements of the UAV, which induce large changes in the visual measurements. In this scenario, a fixed visual covariance matrix will result in large losses of precision.

Next, re-optimization was performed using the new visual covariance obtained from the covariance tuning function. The increases in the unit-weight RMSEs of the IMU measurements and visual reprojections after re-optimization are shown in Figure 7a,b. According to Equation (12) and the aforementioned figures, the IMU residuals become larger after re-optimization; although the visual residuals fluctuate somewhat, they become significantly smaller in some frames. Therefore, the confidence level of the visual measurements was increased by the re-optimization of these image frames. Figure 7c,d show that after re-optimization, the weight of the visual reprojection factor σ^−1^ varies from 100 to 3000. This range corresponds to a reprojection error range of 0.2–3.5 pixels in the image plane. Hence, our resilient covariance tuning-based method for visual–inertial fusion is capable of flexible visual weight adjustments.

Figure 8 compares the numerically downscaled σ^−1^ values shown in Figure 7c to the corresponding IMU unit-weight RMSEs obtained without re-optimization. The tuning of the visual weight factor is correlated with the IMU unit-weight RMSEs, in terms of peak values, and there is also some resemblance in their fluctuations. Therefore, our method can react to changes in IMU precision and apply the appropriate visual weights to make the best use of the sensors’ real-time precision.

### 3.2. Experimental Analysis of the Resilient Covariance Tuning-Based Visual–Inertial Fusion Algorithm

To test the feasibility of the proposed algorithm, we conducted a validation experiment using the entirety of the EuRoc dataset. We also compared the results to those obtained with VINS-Mono without closed-loop detection. The comparison data with the open-source frameworks R-VIO and OKVIS are presented in Table 2 below. Since the accuracy of VINS-Mono is superior to the other two algorithms in most scenarios, in order to facilitate the observation of trajectory characteristics, this paper only selects VINS-Mono for trajectory comparison. The trajectory comparison between VINS-Mono, R-VIO, and OKVIS is described in detail in reference [18]. The validation and algorithm comparison experiments were performed using the Robot Operating System (ROS) suite with the Ubuntu 18.04.6 LTS operating system. The CPU used was an Intel(R) Xeon(R) Silver 4214 running at 2.20 GHz with 48 threads. The results are shown in Figure 9. Our algorithm produced trajectories that were more similar to the ground-truth trajectories than those produced by VIS-Mono. The trajectories produced by our algorithm were much more stable at bends and straights, and their termination points were also much closer to the real termination points.

The absolute pose precision of the proposed algorithm was evaluated using its root-mean-square absolute pose error (RMS-APE) for the EuRoc dataset, as shown in Figure 10. Although large fluctuations occurred in some spots, the RMS-APE only ranged from 0.074 to 0.31 over all of the EuRoc sub-datasets. Furthermore, the localization precision of the algorithm was better than 30 cm across all of the sub-datasets.

The precision of the proposed algorithm was then compared to three other open-source viSLAM frameworks (VINS-Mono, R-VIO, and OKVIS), as shown in Table 2. As the R-VIO algorithm has high computational demands during the initialization stage, it required some time for static initialization. Therefore, the R-VIO data in the table are the best results obtained after multiple trials.

Our algorithm significantly outperformed the VINS-Mono, R-VIO, and OKVIS open-source viSLAM frameworks in terms of precision. Furthermore, our algorithm was the most precise of the algorithms compared over most of the EuRoc sub-datasets, and its precision was better than that of VINS-Mono by 2.39 to 35.65% over the whole EuRoc dataset (18.20% on average).

## 4. Conclusions

In this study, we constructed a covariance-tuning function based on a posteriori unit-weight RMSEs and proposed a method for visual–inertial fusion based on covariance tuning that improves the localization and pose precision of VIO systems. First, nonlinear optimization was performed based on the factor graph model, and the optimized residuals are used to compute the a posteriori unit-weight RMSEs of the visual reprojection. Next, the a posteriori unit-weight RMSEs were passed to the visual covariance tuning function to create a new visual covariance matrix that is used for re-optimization and marginalization. The proposed algorithm was validated over the entire EuRoc dataset and compared to the VINS-Mono, R-VIO, and OKVIS open-source viSLAM frameworks. The results show that our algorithm maintains a high level of pose and localization precision in settings of varying difficulty. The disadvantage of the proposed algorithm is that the introduction of re-optimization may have an impact on the real-time performance of pose calculation, and the influence of gross error factors is not considered when calculating the unit-weight RMSE.

## Figures and Tables

**Figure 1 sensors-22-09836-f001:**
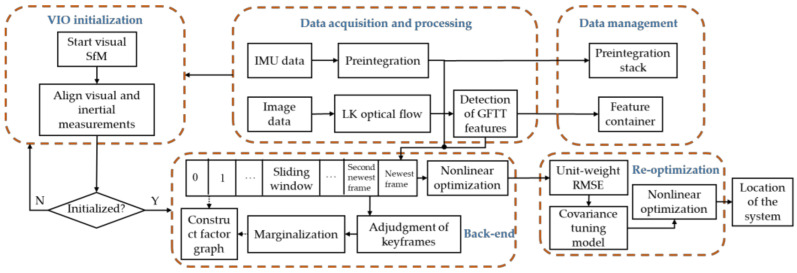
Architecture of the resilient covariance tuning-based visual–inertial fusion algorithm.

**Figure 2 sensors-22-09836-f002:**
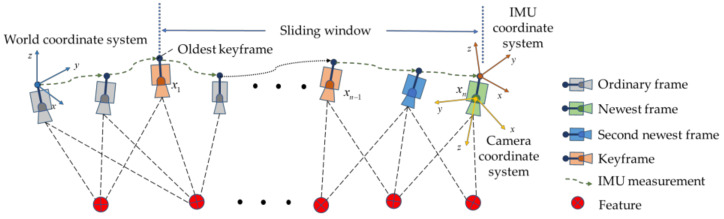
Observation model.

**Figure 3 sensors-22-09836-f003:**
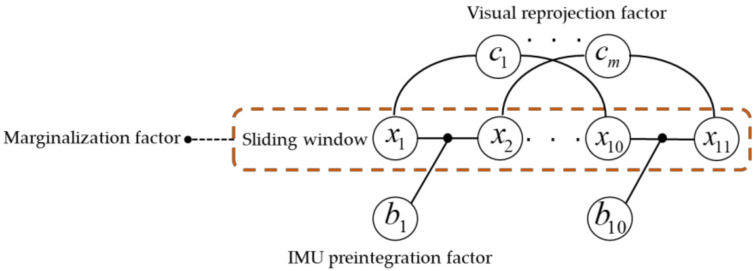
Factor graph model of the algorithm.

**Figure 4 sensors-22-09836-f004:**
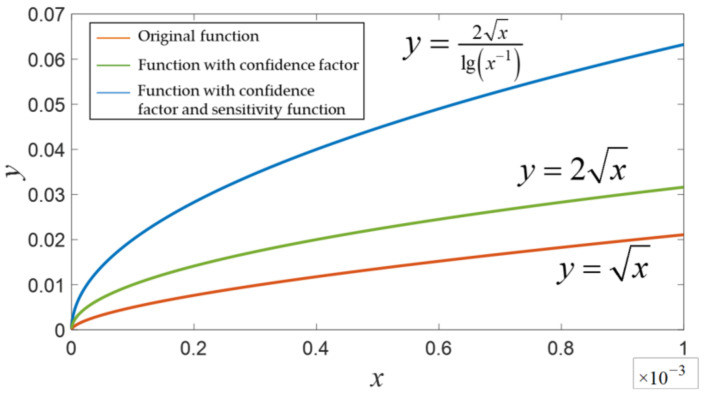
The covariance tuning function.

**Figure 5 sensors-22-09836-f005:**
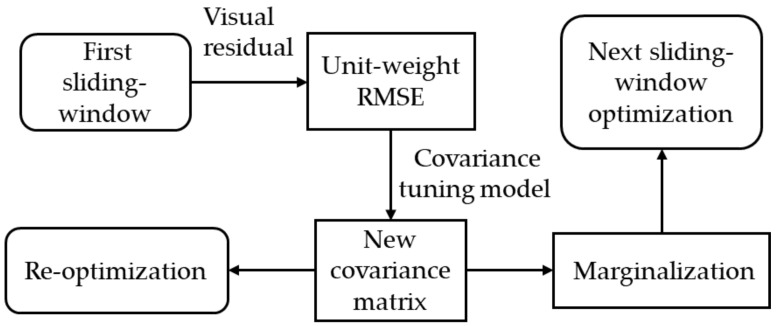
Role of the covariance matrix.

**Figure 6 sensors-22-09836-f006:**
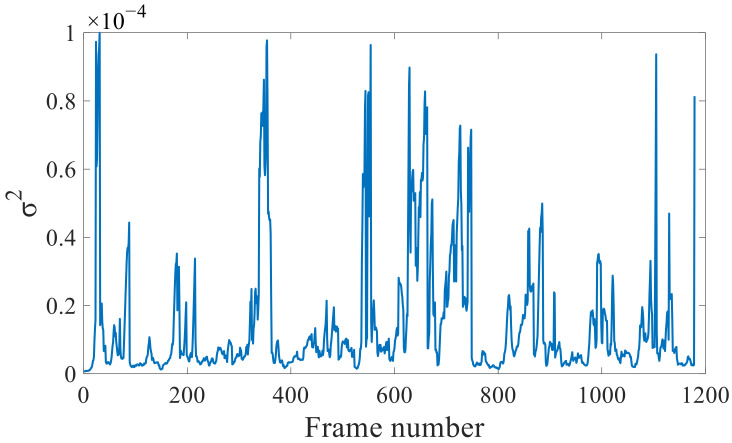
Unit-weight RMSEs of the visual reprojection without re-optimization.

**Figure 7 sensors-22-09836-f007:**
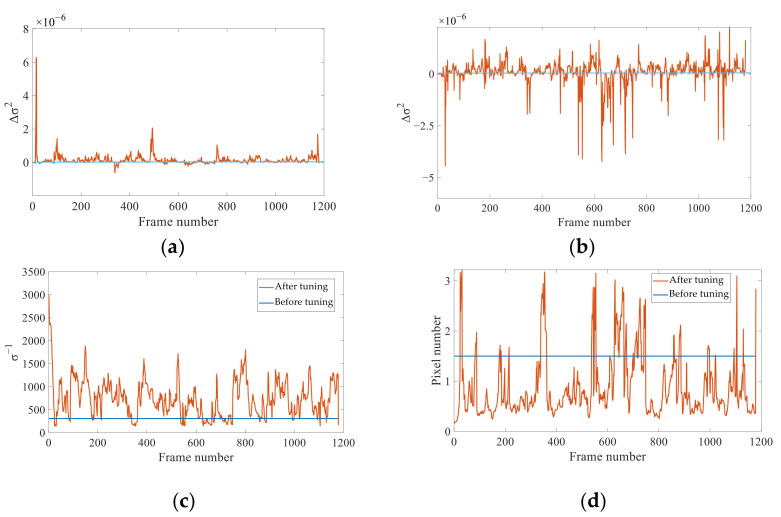
Changes in the relevant parameters after re-optimization. (**a**) Increase in unit-weight RMSE of the IMU measurements. (**b**) Increase in unit-weight RMSE of the visual reprojections. (**c**) Changes in the visual reprojection weights. (**d**) Manifestation of the visual reprojection weights on the image plane.

**Figure 8 sensors-22-09836-f008:**
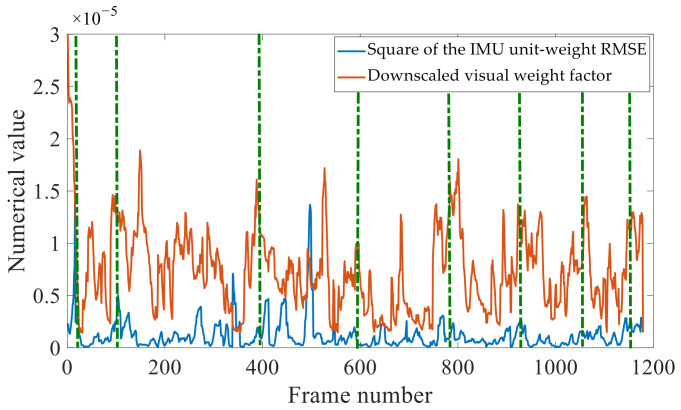
Real-time relationship between visual weight factor adjustments and IMU precision.

**Figure 9 sensors-22-09836-f009:**
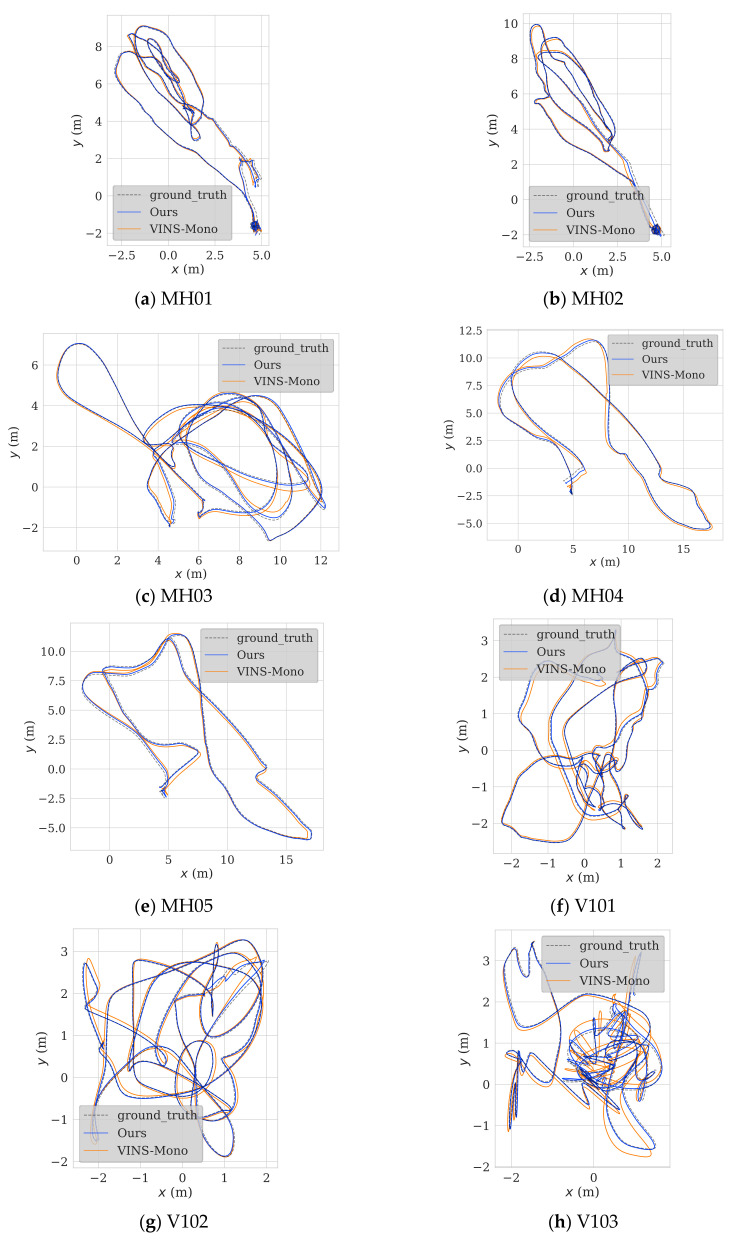
Trajectories computed by the proposed algorithm and VINS-Mono.

**Figure 10 sensors-22-09836-f010:**
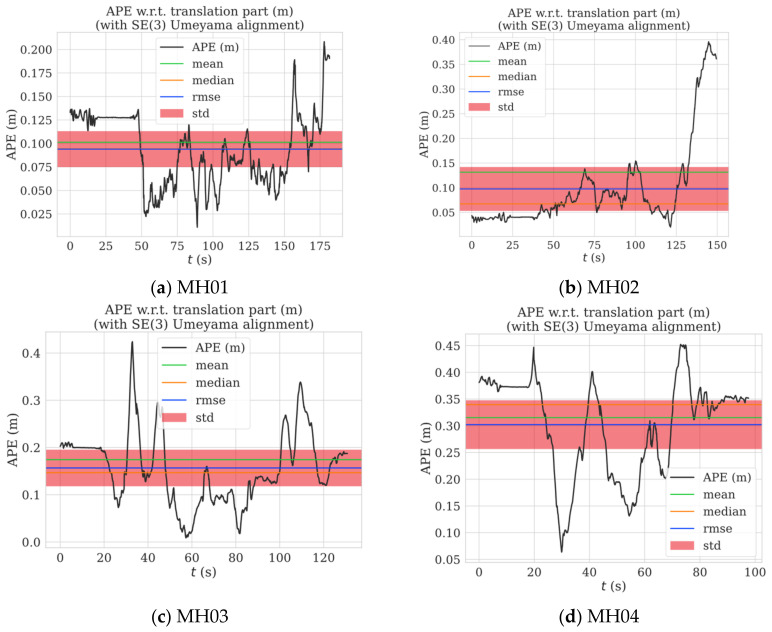
Pose error across the EuRoc dataset.

**Table 1 sensors-22-09836-t001:** Sub-datasets of the EuRoc dataset.

Sub-Dataset	Quantity of Segments	Difficulty	Traveled Distance/m
MH01–MH05	5	E/E/M/D/D	80.6/73.5/130.9/91.7/97.6
V101–V103	3	E/M/D	58.6/75.9/79.0
V201–V203	3	E/M/D	36.5/83.2/86.1

**Table 2 sensors-22-09836-t002:** Comparison between the proposed algorithm and several open-source frameworks in terms of APE. Bold indicates the excellent value of the results in the same row.

Sub-Dataset	Difficulty	Our Algorithm	VINS-Mono	R-VIO	OKVIS	Improvement in Precision Compared to VINS-Mono
MH01	Easy	**0.101233**	0.157314	0.328240	0.331345	35.65%
MH02	Easy	**0.131429**	0.178440	0.639892	0.387684	26.35%
MH03	Medium	**0.174250**	0.195266	0.233700	0.268468	10.76%
MH04	Hard	0.315295	0.439647	1.297599	**0.287485**	28.28%
MH05	Hard	**0.221533**	0.303964	0.521598	0.393153	27.12%
V101	Easy	**0.079860**	0.088830	0.098709	0.095340	10.10%
V102	Medium	**0.106666**	0.111855	0.134505	0.148746	4.64%
V103	Hard	0.159576	0.187750	**0.151586**	0.211350	15.01%
V201	Easy	**0.074389**	0.094752	0.123188	0.099128	21.49%
V202	Medium	**0.137455**	0.168498	0.169666	0.176457	18.42%
V203	Hard	0.280010	0.286872	0.837517	**0.237462**	2.39%

## Data Availability

Publicly available datasets, EuRoc, were analyzed in this study. This data can be found here: https://projects.asl.ethz.ch/datasets/doku.php?id=kmavvisualinertialdatasets (accessed on 17 July 2022).

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
