# Peer review of "A Resilient Method for Visual–Inertial Fusion Based on Covariance Tuning"

_sensors, 2022, doi:10.3390/s22249836_

Round 1
Reviewer 1 Report
Congratulations to the authors of a very interesting article.
I only have editorial comments to it, which in my opinion should be taken into account before its final publication, i.e .:
The lowercase x in formula (1) represents a vector, so it should be bold.
The same is true for the velocity vector on line 154 and the variables described below (on lines 150, 152).
Additionally, the variables representing the vectors should not be italicized - this note applies to the entire text of the article.
I recommend a detailed review of the manuscript in terms of the uniform application of vector designations, e.g. in formula (1), vector elements are separated by a comma and not in formula (6).
The first concerns the introduction in which the genesis of the problem can be described in more detail in terms of the innovative nature of the proposed solution. The second one concerns conclusions which could be more detailed and could also refer to the disadvantages of the proposed solution.
Author Response
Dear reviewer:
Thank you for your comments and suggestions on our paper! We quite agree with you. Therefore, we made modifications to our article. You can see them in the modified and uploaded manuscript, and we have replied to your comments one by one in the attachment "cover_letter1" that we submitted to you.
Please see the attachment.

Reviewer 2 Report
The manuscript deals with the design of a localization algorithm based on the weighted fusion of visual and inertial information. To improve this already known approach it tries to tune the weights for these parts of information. The proposed algorithm was compared to other selected methods on a benchmark dataset. The manuscript is well structured and basic notions are explained but it still need an experienced reader to really understand all details of the method. Experiments have proved the new method outperforms other selected methods and it can be beneficial for the praxis. However, I have some comments to improve its quality.
1. In recent years many papers were published in the area of both localization approaches in the SLAM area. However the total number of 18 citations is maybe satisfactory for a conference but not for a journal paper. A broader and more detailed state-of-the-art must be elaborated to be able to explain the sense of such a kind of algorithms as well as the choice of selected methods that could be comparably strong for the competition with the proposed one.
2. Also some MDPI citations, mainly of papers from this journal would be welcome. Why do you want to publish just in Sensors?
3. Please, consider larger letters in the figures.
4. Please, explain all abbreviations used in the text. A table of abbreviations is missing.
5. Fig. 1 is the core of your proposal. Therefore, it must be described in detail, each block and each notion.
Author Response
Dear reviewer:
Thank you for your comments and suggestions on our paper! We quite agree with you. Therefore, we made modifications to our article. You can see them in the modified and uploaded manuscript, and we have replied to your comments one by one in the document "cover_letter2" that we submitted to you.
Please see the attachment.

Reviewer 3 Report
This paper proposes an elastic viSLAM algorithm based on covariance adjustment. A covariance tuning function is constructed to optimize the weight ratio of visual observation information, which improves the performance of VI localization algorithm in complex environments, and the performance is verified by measured data. The overall logic of the paper is clear and the content is detailed. The methods and results are of certain reference significance for the subsequent research, but there are still the following problems, which should be explained and modified.
1. Why is GFTT feature used? The VMS-MonO framework considers the real-time performance of the algorithm and uses relatively simple feature point types. How does GFTT compare with the feature point types used by VMS-MonO in terms of real-time performance? Is it helpful to improve accuracy?
2. The size of Table 1 is too large, should be modified;
3. The horizontal coordinate "Frame number" in Fig. 5, 6 and 7 is cryptic, needs to be explained clearly;
4. Fig. 9, included the trajectory is not clear enough, and the font is too small to see the coordinate scale information, needs to be modified;
5. Has the performance of variance tuning function been demonstrated in relevant literature? If so, it should also be explained in detail?
6. Fig. 5 is not clear enough, the location and size of the picture should be noted to ensure readability and beauty;
7. Fig. 8 only compares the VMS-Mono trajectory. In fact, other two algorithms are also mentioned in the conclusion of the paper, so the corresponding charts should be added.
8. The correct format of references should be confirmed, such as reference 10, etc.
Author Response
Dear reviewer:
Thank you for your comments and suggestions on our paper! We quite agree with you. Therefore, we made modifications to our article. You can see them in the modified and uploaded manuscript, and we have replied to your comments one by one in the attachment "cover_letter3" that we submitted to you.
Please see the attachment.

Round 2
Reviewer 2 Report
All comments were properly considered and the quality of the manuscript has been improved to a level that enables its publishing.
Reviewer 3 Report
Good job, I have no more comments.